# Effects of the Expression of Random Sequence Clones on Growth and Transcriptome Regulation in *Escherichia coli*

**DOI:** 10.3390/genes13010053

**Published:** 2021-12-24

**Authors:** Devika Bhave, Diethard Tautz

**Affiliations:** Max Planck Institute for Evolutionary Biology, August-Thienemann-Str. 2, 24306 Plön, Germany; bhave@evolbio.mpg.de

**Keywords:** de novo gene evolution, random sequences, fitness, *E. coli*, transcriptome

## Abstract

Comparative genomic analyses have provided evidence that new genetic functions can emerge out of random nucleotide sequences. Here, we apply a direct experimental approach to study the effects of plasmids harboring random sequence inserts under the control of an inducible promoter. Based on data from previously described experiments dealing with the growth of clones within whole libraries, we extracted specific clones that had shown either negative, neutral or positive effects on relative cell growth. We analyzed these individually with respect to growth characteristics and the impact on the transcriptome. We find that candidate clones for negative peptides lead to growth arrest by eliciting a general stress response. Overexpression of positive clones, on the other hand, does not change the exponential growth rates of hosts, and they show a growth advantage over a neutral clone when tested in direct competition experiments. Transcriptomic changes in positive clones are relatively moderate and specific to each clone. We conclude from our experiments that random sequence peptides are indeed a suitable source for the de novo evolution of genetic functions.

## 1. Introduction

The origin of novelty is a fundamental theme in evolutionary genetics. While the focus is often on changes in existing genes and their effects on the downstream pathways, comparative genomic analysis has shown that the de novo evolution of new genes also contributes much to evolutionary innovations (reviewed in [1,2,3,4]). Different mechanisms are responsible for generating new genes, including segmental duplications, retrogene insertions, and de novo evolution from non-coding DNA. While the latter mechanism has initially been considered to be very unlikely (see review in [5]), it has been documented for several well-studied cases (see [6,7,8] for recent reviews). De novo genes can emerge out of previously non-coding regions. In eukaryotes, these are mostly the intergenic region, which are transcribed by flanking or spurious promotors [9]. But non-coding regions are also abundant in compact genomes, such as viruses or bacteria in the form of alternative reading frames of coding regions. There are several well documented cases where genes evolved de novo within an existing gene, both in viruses [10], as well as in bacteria [11,12,13].

Nonetheless, how frequently such a de novo emergence could occur remains a matter of active debate. A key question in this debate is: what fraction of random sequence open reading frames (ORFs) have the potential to interact with intracellular components in a way to allow them to be subjected to positive selection, and thus to establish a novel function for an organism?

We have previously addressed this question by expressing libraries with random sequence ORFs in *Escherichia coli* and studying frequency changes of each clone while allowing growth under competitive conditions for several generations in four growth cycles [14]. We found that about 70% of the total clones showed a consistent positive or negative effect on the growth across multiple experimental replicates. However, the results of this experiment were challenged on the basis of the choice of an expression vector, which could have potentially interfered with the interpretation of the frequency changes of clones owing to the vector-specific negative effect on the host [15,16]. It was argued that the empty vector drives a strong expression after induction and also produces a small 38 amino-acid long peptide encoded by the multiple cloning site (MCS), either of which can be detrimental for the host (see Appendix A for a scheme of the vector design). By inserting random sequences into the MCS, it is possible to either suppress the generally harmful effects of overexpression or, more specifically, the potentially harmful expression of the peptides [16]. Either way, this would only impact the clones that showed a positive growth effect since they can be interpreted to alleviate the negative effect of the empty vector non-specifically. However, the overall effect sizes of several of these clones would argue against such an interpretation [17].

Based on a re-examination of the data using a new analysis pipeline that included not only the full-length peptides as in [14] but also all shorter versions of the peptides that were in the library, we found no evidence of a strong negative vector effect in the experiments [18]. The study showed that approximately 36% of peptides have a negative effect on cell growth (called NEG_Pep in the following), 48% had no significant effect (NS_Pep) and 16% of peptides had a positive effect on the relative growth of the host (POS_Pep). Interestingly, this study showed that shorter sequences (between 8–20 amino acids) constitute the highest fraction of positive peptides after four growth cycles. However, given that these experiments were done in the context of the entire clone library, clonal interference can play a significant role in affecting the growth trajectories of individual clones [19,20].

Therefore, a key for a better understanding of the effects of random sequence ORF expression in *E. coli* is to trace specific effects of individual clones. Here we study a set of such candidate ORFs derived from the original library. We test individual growth phenotypes and monitor transcriptome changes after the induction of expression to get insights into their effects on the cells.

We find that the expression of the negative clones induces a relatively generic stress response in the host. In contrast, the positive clones show no stress response but clone-specific effects, and growth advantage in direct competition with an NS_Pep clone. Hence, our data support the notion that a pool of random sequences can easily provide the raw material for the de novo evolution of genes.

## 2. Materials and Methods

### 2.1. Strains, Plasmid and Growth Conditions

Three strains, *E. coli* K-12 DH10B (NEB^®^ 10-β competent, NEB catalogue #C3019H), *E. coli* B REL606 [21], and REL607 were used as backgrounds for this study. A multicopy expression vector, pFLAG-CTC™ (Sigma-Aldrich #E8408, Sigma-Aldrich, St. Louis, MO, USA), was used for the cloning and expression of candidate ORFs. Glycerol stocks were made by adding 700 µL of fully grown cultures into 700 µL of 50% glycerol and stored at −70 °C. Liquid media used for growth were: Lysogeny broth (LB) Lennox containing 10 g/L tryptone, 5 g/L yeast extract and 5 g/L NaCl, or Minimal medium (M9-Glucose) containing 33.9 g/L NaH_2_PO_4_, 15 g/L of KH_2_PO_4_, 2.5 g/L NaCl, 5 g of 1.8 M NH_4_Cl, 50 μL 1 M CaCl_2_·6H_2_O, 1 mL 1 M MgSO_4_·7H_2_O, and 10 mL 20% glucose. M9 media components were autoclaved and added separately to prevent precipitation and charring of glucose. The revival was done by streaking on agar plates with appropriate media (generally LB agar or M9 Glucose agar) supplied with 50 µg/mL Ampicillin (selection marker for plasmid) to obtain single isolated colonies that serve as clones for experimental replicates. Bacteria were generally incubated overnight for 16–18 h at 37 °C, shaking at a speed of 250 RPM (if shaking), unless otherwise mentioned.

### 2.2. Cloning of Selected Candidates into E. coli Strains

Selected candidate sequences were pulled out from the initial random library through PCR. Specific primers were designed for each sequence of interest (list provided in Appendix A), which were later used to amplify from the stored library plasmid DNA. Sequences were amplified using a 2-step Phusion^TM^ High-Fidelity PCR Master Mix (Thermo Fisher Scientific) with the following PCR conditions: initial denaturation at 98 °C for 30 s, followed by 30 cycles with annealing at 72 °C for 20 s, and denaturation at 98 °C for 10 s. The final extension was performed at 72 °C for 10 min, followed by cooling at 8–12 °C. Phusion^®^ PCR kit uses a high-fidelity Phusion polymerase with a 5′ to 3′ polymerase and 3′ to 5′ exonuclease activity. The amplified products were purified using a QIAquick PCR purification kit (Qiagen, Hilden, Germany), then used for downstream cloning. The purified amplicons and purified vector DNA were digested using HindIII-HF™ and SalI-HF™ restriction enzymes for 1 h at 37 °C, followed by purification using a QIAquick PCR purification kit. Purified products were ligated with 1 μL T4 DNA ligase (protocol as per NEB^®^) for 10 min at room temperature (benchtop) using a 3:1 insert to vector ratio. Ligation products were transformed in already competent cells background strains using chemical transformation. Commercial competent cells were used for the K-12 DH10B strain (NEB^®^ 10-β high efficiency). Transformation of B REL606 and B REL607 was achieved via the chemical competence method. For this, cells were prepared by growing cultures overnight in 4 mL LB medium and inoculating 500 μL of the pre-culture into fresh 200 mL LB medium and allowed to grow at 37 °C, 250 rpm until an OD_600_ of 0.45–0.55 was reached. The culture was collected in four 50 mL FalconTM tubes and centrifuged for 10 min at 4°C, 3000 rpm. Pellets were gently resuspended in 1 mL chilled TBF-I solution (30 mM KOAc, 100 mM RbCl, 50 mM MnCl_2_ and 10 mM CaCl_2_) and filled up to 15 mL with the same. Tubes were incubated on ice for 1 h followed by centrifugation for 10 min at 4 °C, 3000 rpm. Pellets were then resuspended in 4 mL TBF-II solution (10 mM MOPS, 10 mM RbCl, 75 mM CaCl_2_ and 15% Glycerol).

Several transformation positive clones were freshly inoculated in 4 mL LB+ Amp media to prepare glycerol stocks the next day. The same colonies were also used for colony PCR to confirm the presence of insert. Colony PCR was done by taking a part of a fully-grown colony from the transformation positive plate and resuspending in 10 μL of sterile water. The suspension was heated at 98 °C for 10 min and used as a template for subsequent 2-step Phusion PCR using specific primers.

For confirmation of clones, pure amplicons were used for Sanger sequencing [22]. Common outer primers were used to amplify inserts (see Appendix A). Output sequence files were analyzed using Geneious prime (version 2019.1.3 or later) and CodonCode Aligner (version 7.0.1).

### 2.3. Growth Measurements

All growth curve measurements were performed on a Tecan M nano+ (Tecan Deutschland GMBH, Crailsheim, Germany) plate reader. Cultures from frozen stocks were streaked on appropriate media plates with ampicillin (50 μg/mL) and incubated overnight at 37 °C (unless otherwise mentioned). The next day, single colonies were inoculated in 200 µL of either LB or M9 medium with ampicillin in 96 welled plates and incubated with vigorous shaking for 16–18 h. The following day, growth curves were set up by adding 2 µL of overnight culture to 200 µL of medium with ampicillin with or without 1 mM isopropyl β-D-1-thiogalactopyranoside (IPTG) for induction of expression. Growth curves were usually recorded for 15–24 h with 5 min of orbital shaking before reading OD_600_ every 10 min at the desired temperature. The manual time series experiment was performed to obtain the colony-forming units in growing cultures. OD_600_ measurements in the plate readers measure turbidity and can be overestimated by factors like exopolysaccharide production. Strains were revived by standard procedures and growth measurements were started in larger volumes of 5 mL with a starting dilution of 1:100. After every hour of growth with or without IPTG induction at 250 rpm, cultures were plated on LA+Amp plates at an appropriate dilution and incubated overnight. Colonies were counted using a colony counter and CFU/mL at each time point was calculated. 

### 2.4. Population Size Estimation Using Manual CFU Counts

Effects of NEG_Peps on the host viability were tested by growth estimation via CFU counts. Candidate strains were streaked on LA Ampicillin plates from respective glycerol stocks and incubated overnight at 37 °C. The next day, three single colonies were inoculated in 4 mL LB Ampicillin and incubated at 37 °C at 250 rpm overnight (16 h). The following day, 40 µL from the overnight culture was inoculated into 4 mL LB Ampicillin tubes (1 in 100 dilution). Cells were allowed to grow for about 3–4 h until the OD_600_ reached ~0.4 (exponential phase). The uninduced time point (T0) was plated at an appropriate dilution to obtain a range of 30–300 colonies on LA ampicillin plates. Subsequently, the growing cultures were induced by adding 1 mM IPTG to activate the NEG_Pep expression, and cells were plated at appropriate dilutions after 5 min, 15 min, 30 min, 1 h, 6 h and 10 h of induction. The observed colony counts were used to estimate CFU/mL at each of the above-mentioned time points.

### 2.5. Competition Experiments

Competitive fitness assays were performed using two *E. coli* strain backgrounds: B REL606 and B REL607 as described previously [23]. REL606 and REL607 are the ancestral strains of the *E. coli* long-term evolution experiment, which differ by a single point mutation in the arabinose utilization (araA) gene of REL607, which allows it to metabolize arabinose. The two competitors can be distinguished by their arabinose utilization phenotypes; (REL606) Ara^-^ and (REL607) Ara^+^ that produce red and white colonies respectively on Tetrazolium agar (TA) indicator plates. The two backgrounds were transformed with our candidate clones on the pFLAG vector. First, strains were streaked on M9 Glucose Ampicillin agar plates and incubated at 37 °C to obtain single colonies. Then, 4–8 colonies were inoculated the next day in 4 mL M9 Glucose Amp media per competitor and allowed to grow at 37 °C for 18 h, shaking at 250 rpm. On the subsequent day, a 1:1 volume of overnight cultures were used from each competitor strain and mixed thoroughly. After mixing them in a 1:1 ratio, 40 μL of the mixture was inoculated into fresh 4 mL M9 Glucose+ Amp media with IPTG. Simultaneously, the mixture was plated (T0) on TA+Amp plates at an appropriate dilution such that the colony count was between 30–200 (statistical significance range) and incubated at 37 °C overnight. After 24 h of growth, the competition cultures were plated at an appropriate dilution and incubated at 37 °C overnight. For 48-h competitions, 40 µL culture from the 24-h tubes was transferred into fresh 4 mL media and allowed to grow for the next 24 h at the same growth conditions. Colonies from the initial plating were counted and recorded as T0. Colonies plated after 24 h cycle were counted, and the relative fitness of strains was determined.

Relative fitness was calculated as described in [23,24]:$$mA=\;\ln\left(A_{f}/A_{i}\right)$$
$$mB=\;\ln\left(B_{f}/B_{i}\right)$$
W = mA/mB 
where A and B are the two competing strains, i and f are the initial and final population densities (CFU/mL) of each competitor, and W is the relative fitness of strain A w.r.t B. Ln is the natural logarithm. Malthusian parameter (m) of a strain (A or B) reflects population density changes over time.

### 2.6. Total RNA Extraction

Cultures (5 mL) were prepared from single colonies in triplicates. Inoculated tubes were allowed to grow until they reached the exponential phase OD_600_ of 0.4–0.5. At this point, 1 mM IPTG was added, and cultures were further incubated for 1 h with shaking. Aliquots of 500 μL were taken and centrifuged at 13,000 rpm. After centrifugation, they were vigorously (with Vortex shaker) resuspended in 1 mL of RNAprotect Bacteria Reagent (Qiagen Catalogue # 76104). The mixtures were incubated for 5 min at room temperature and then centrifuged for 10 min at 5000× *g*. Supernatants were decanted, and pellets were stored at −20 °C for up to one week. Total RNA extraction was performed using the RNeasy^®^ Mini kit (Qiagen, Catalogue #74106) following the kit protocol. Final elution was performed with 40 μL of pure water. Total RNA samples were stored at −70 °C until further use.

### 2.7. Hybridization and Feature Extraction Using E. coli Microarray Chips

RNA labeling and microarray hybridization were performed by following the supplier’s protocol of the chips (Agilent, Santa Clara, CA, USA). The labeling kit generated cyanine labeled cRNAs which were amplified using the WT kit primer mix (mixture of oligo dT and random nucleotide-based T7 promoter primers), generating cRNA from samples. The provided spike-in controls were also labeled and amplified with the samples. Labelling, hybridization, washing and scanning were performed using the standard protocol from the Agilent user guide (Low Input Quick Amp WT labeling kit, Catalogue #5190-2943). We performed one-color microarrays with the commercially available Agilent *E. coli* microarrays 8x15K, P/N G4813A, design ID 020097 with complete gene probes list. All experiments were run in triplicates.

### 2.8. Differential Expression Analysis

We used the Limma software package on R [25] to analyze the microarray data. All analyses are based on comparing expression level differences to RNA from the induced pFLAG empty expression vector. The results for the log2 fold changes (logFC) compared to the vector are provided in Appendix A for the NEG_Pep clones and in Appendix A for POS_Pep clones. Top genes with logFC expression >1 or <−1 were extracted from the respective tables. Gene Ontology (GO) analysis was done using Panther (version 16.0) [26].

## 3. Results

Individual candidate clones in the present study were selected from the random sequence library described in [14] and based on the re-analysis of the data described in [18] (see also introduction for the results from this study). We chose clones independent of their sequence characteristics but representing the spread of log2fold changes after growth cycle 4. Six clones each with positive (POS_Pep) and negative (NEG_Pep) effects on cell growth were chosen, plus five clones that had shown no significant (NS_Pep) growth change. All clones code for full-length peptides (65 amino acids) except POS_Pep4b, 6 and 7, which have a premature stop codon, and hence produce shorter peptides with 16, 28 and 45 amino acids, respectively (see Appendix A for the sequences). Figure 1a depicts these clones in the overall cloud of clones for the different growth cycle comparisons. We find that the empty vector, which expresses a 38 amino-acid long polypeptide, shows only a small non-significant negative frequency change, i.e., at least in the bulk experiment, it does not have a strong negative effect on its own (Figure 1—pFLAG).

Figure 1b shows the individual growth trajectories across the four cycles for all candidate clones and the pFLAG vector. The NEG_Pep clones show a fast drop in the first cycle. Four of the POS_Pep clones show an exponential increase in abundance across the four cycles, while POS_Pep2 and POS_Pep3 show a linear increase. These differences suggest that growth trajectories may be subject to sequence-specific influences. NS_Pep abundances and the pFLAG vector fluctuate up and down in the four cycles, i.e., show no clear trend (Figure 1b).

### 3.1. Growth Characteristics of NEG_Pep Clones in E. coli

We first focused on the growth characteristics of the negative clones. Growth curves were recorded for each clone under induced and non-induced conditions. All six clones show an initial growth delay after induction of the peptide before they resume growth (Figure 2a). The duration of the lag times differs slightly between the different clones, with NEG_Pep3 and NEG_Pep4 being particularly variable between replicates (Figure 2b). Growth resumes after the lag, but the rates remain significantly lower than the respective non-induced controls for five out of the six candidate clones (Figure 2c).

The observed lag phase could be caused by cells dying after the induction of peptide expression and only cells that develop a resistance mutation would resume growth after some time. To assess this possibility, we determined viable cell numbers to better understand the underlying cause of the growth delay. For this, we grew the bacteria until the log phase before adding IPTG. Samples were then taken at different times (up to 10 h) and were plated on LB Ampicillin plates without IPTG. Figure 3 shows that viable cell numbers do not go down much in the first hour after IPTG induction, indicating that the cells are getting only growth-arrested but not immediately killed by the IPTG induction. However, viable cell numbers drop by two orders of magnitude by 10 h (Figure 3). For NEG_Pep5 and NEG_Pep6 we see a slight increase between 6 h to 10 h, indicating that they are recovering during this phase, although the experimental conditions are somewhat different than in the experiments shown in Figure 2.

To assess whether the negative effects of these clones are due to the expressed protein or mRNA, we tested six corresponding in-frame STOP codon versions of the clones that should express only the first three vector-encoded amino acids (Appendix A). We found that four out of the six NEG_Pep clones still showed a growth delay due to the expressed RNA (Appendix A), implying that the RNA alone can negatively impact the growth.

### 3.2. Transcriptomic Response to NEG_Pep Expression

Monitoring RNA expression changes after IPTG induction is a way to study the host cellular response to random ORF production. We used standardized *E. coli* microarrays to assess gene expression changes (Agilent). Cells were grown to log phase and then induced with IPTG. RNA was harvested 1 h after induction (i.e., when most cells are arrested but viable—see above). Cells carrying the induced vector plasmid without the insert were used as controls for comparing expression changes (see the summary of expression changes for all genes in Appendix A).

The overall expression comparison via a Principal Component Analysis (PCA) indicates that each clone and their corresponding STOP codon versions show unique responses, whereby the main differences along PC1 correlate with peptide expression versus RNA expression (Figure 4a). Given that there were a large number of genes with significant changes (Figure 4b), especially for the clones expressing the respective peptides, we decided to focus on the more detailed analysis on the genes with the highest fold-change values (Figure 4b). Heatmap (based on log2 fold change) of these genes shows an apparent clustering of NEG peptides versus the STOP codon versions of the clones (Figure 4c). This distinct picture suggests very different transcriptome reactions for expressing the proteins versus RNAs for these clones. But apart from these significant differences, it is also clear from the heatmap that there are some distinct clone-specific differences.

We used the gene lists of the top over-expressed genes to retrieve GO enrichment information from Panther GO [27,28]. The top enriched categories overlap strongly for all six candidates (Figure 5). They include, in particular, stress response genes in various combinations, including protein folding. Only NEG_Pep3 shows fewer of these genes enriched. None of the stop codon versions of the peptides shows enrichment for these stress response genes; two show no enrichment for any GO category, one (NEG_Pep1_STOP) for “galactitol metabolic process” and the others for efflux transport processes (Figure 5). We conclude that a major consequence of the expression of the six NEG peptides is a strong stress response, with some modulation (e.g., in the case of NEG_Pep3). Furthermore, the stop codon versions of the respective candidate clones do not elicit the stress response. However, at least some of them cause almost equal growth inhibition as their corresponding peptide expressing clones (Appendix A), substantiating the notion that the RNA itself can cause effects on growth.

### 3.3. Growth Characteristics of POS_Pep Clones

In the second set of experiments, we focused on the analysis of the POS_Pep clones. First, we assessed the growth characteristics in comparison to the NS_Pep clones. Growth curves were recorded for each clone under induced versus non-induced conditions. Growth rates were determined during the exponential phase, but we did not find significant growth differences between the non-induced and induced conditions for any of them (Appendix A).

However, measurement of maximum growth rates of cultures during the exponential phase provides only a proxy for fitness. To increase the sensitivity of detection of growth differences, we performed competitive fitness assays in two strain backgrounds (*E. coli* B REL606 and *E. coli* B REL607) using the red-white selection as described in [23]. The competitive fitness was calculated by estimating the population densities (CFU/mL) of each competitor at the beginning (T0) and the end (two 24 h growth cycles) of the competition. Relative fitness was calculated using Malthusian parameters of each competitor as described before (see Methods). Genotypes with higher fitness produce more descendants and consequently outcompete their less-fit competitors.

For the competitive fitness assays, each competitor was mixed in a one-to-one ratio from a fully-grown overnight culture (~10^9^ cells/mL) and allowed to compete in IPTG supplemented minimal medium (M9+Glucose) for two 24 h cycles. Note that competitive fitness assays measure the colony numbers as opposed to the growth rates. Swapped background strains were also engineered and tested to eliminate background-related effects.

To avoid the problem of a possible negative effect caused by the peptide expressed by the empty vector [16], we used one of the NS clones (NS_Pep3) in the comparative analysis of growth rates in the two strain backgrounds (see Appendix A to support this choice). Most peptides identified as positive in the overall analysis also show a fitness advantage in the competition experiments (Figure 6). However, the effects are weak for POS_Pep1 and POS_Pep7, although both are very competitive in the bulk experiments (Figure 1). This suggests that the experimental conditions (minimal medium versus LB and two cycles versus four cycles) influence the performance of the clones. Overall, however, the experiment supports the notion that random peptide expression can lead to a growth advantage for the respective hosts.

### 3.4. Transcriptomic Response to POS_Pep Expression

Using the above-described microarray system, we have also studied the transcriptomic responses in cells that express POS_Pep clones. These show a very different pattern compared to the NEG_Pep clones. We generally observe a much weaker response at the transcriptional level (fewer genes and lower log2 FC values—see the summary of expression changes for all genes in Appendix A), and no activation of the general stress genes typical for the NEG_Pep clones (see above). Further, there is no apparent commonality of transcriptomic responses between the clones; each shows different top GO categories for the relatively small sets of genes with log2 FC over-or under-expression (Table 1), including two without any GO enrichment (POS_Pep2 and POS_Pep7).

## 4. Discussion

Studying the effects of expression of random sequence ORFs in cells can be considered a proxy towards understanding the evolution of new gene functions out of non-coding sequences [14,15]. Here we have focused on a subset of candidate clones that were previously found to represent negative (NEG_Pep), non-significant (NS_Pep) and positive (POS_Pep) peptides when tested in the background of the growth of the whole library.

We find that expressing the NEG_Pep clones causes a growth arrest of the cells and a strong upregulation of stress response genes. Similar stress responses are also known from clones that are used for recombinant protein production purposes [29,30]. Stress response genes have evolved to allow bacteria to cope with different types of stress conditions elicited by changing environments [31]. We find here that their activation through the induction of NEG_Pep expression leads to an arrest in cell growth and eventually the death of a large number of cells. However, the surviving cells recover from this arrest after up to 10 h in culture and return to almost normal growth. While we did not further explore how this switch back to growth is achieved, it can explain why the clones are not entirely lost in the bulk experiments with the whole library. Although the activation of the stress response is a common theme among these clones, we see also some more specific effects. NEG_Pep3 and NEG_Pep4, for example, show additional sets of differentially regulated genes in response to IPTG induction. This suggests that there are, besides the generic responses, also some clone-specific responses, depending on the expressed peptide.

Our experiments with the in-frame STOP codon versions of the NEG clones provide insights into the effect of RNA versus protein in these random sequence clones. For two clones, we find a complete recovery of the expected growth; for another two clones, we see only a partial recovery, and the final two clones still show a similar lag as their corresponding peptide expressing versions (Appendix A). Interestingly, these latter two clones (Neg_Pep5_STOP and NEG_Pep6_STOP) do not elicit the generic stress response of the transcriptome (Figure 4c), implying that another pathway mediates their effect on growth delay. This observation also indicates that it is not the over-expression that inhibits the host growth per se, but that there are RNA and protein-specific effects. The fact that growth delay can also be caused by RNA transcripts alone, i.e., independent of translation, was also found for clones used for recombinant protein production [32] and in experiments expressing different GFP RNA variants [33].

In contrast to the NEG_Pep clones, the POS_Pep clones do not affect the growth rates after induction. Furthermore, in direct competition with a NS_Pep clone, they show a competitive growth advantage. The transcriptomic response of POS_Pep clones is also different from the one we find for the NEG_Pep clones. Only a few genes are upregulated or repressed, and there is no common theme among the GO categories for the enriched genes. In independent repetition experiments with slightly changed conditions, we have noted that even these responses were not stable and that different sets of genes came up in the top lists for the POS_Pep clones. Hence, we conclude that their effects are unlikely to be directly at the transcriptional level. The changes that we observe in the transcriptome may be secondary effects of interactions of these peptides within the cytoplasm.

The NS_Pep clones do not elicit a growth arrest when induced but can show a lowered growth rate, depending on growth conditions and host background (Appendix A). This could also explain why their abundance fluctuates between cycles, given that the overall-changing clone compositions during the cycles might create somewhat different conditions in each cycle.

## 5. Conclusions

Our analysis confirms the overall notion that random sequences can serve as raw material for new evolutionary adaptations. While peptides or RNAs with deleterious effects would be quickly purged from a population, sequences with neutral or initially positive effects could persist in the population and could eventually become optimized by acquiring additional beneficial mutations. Whether a given sequence can provide a growth advantage will necessarily depend on the environmental conditions. For example, Knopp and colleagues [34] have studied whether random sequence peptides can provide antibiotic resistance and found several clones with such an effect. Functional analysis showed that the effect is caused via membrane depolarization, which decreases aminoglycoside uptake and thus provides a growth advantage under the presence of an antibiotic. However, when these clones were competed against the empty vector plasmid, they showed a growth disadvantage [34], i.e., they would be categorized as negative in our comparisons. Hence, any growth advantages or disadvantages triggered by random sequence clones need to be seen in the context of the competitors in the experiment [15].

It would seem likely that growth differences measured in the bulk experiments with many competing clones would not be directly comparable to growth differences when only two specific clones compete. Hence, the 1:1 competition experiments that we have performed provide only a proxy to what might happen during growth in a complex library with many competitors. After all, with a third of clones that drop quickly in frequency due to strong negative effects on the cells, the remaining clones would necessarily rise in relative frequency. It would therefore seem possible that the POS_Pep clones are simply the least deleterious ones that can grow better when the NEG_Pep clones decrease in cell abundance. In the future, it will therefore be important to test libraries that are composed of subsets of clones that exclude the NEG_Pep clones, and then to trace the individual growth trajectories of the remaining clones.

## Figures and Tables

**Figure 1 genes-13-00053-f001:**
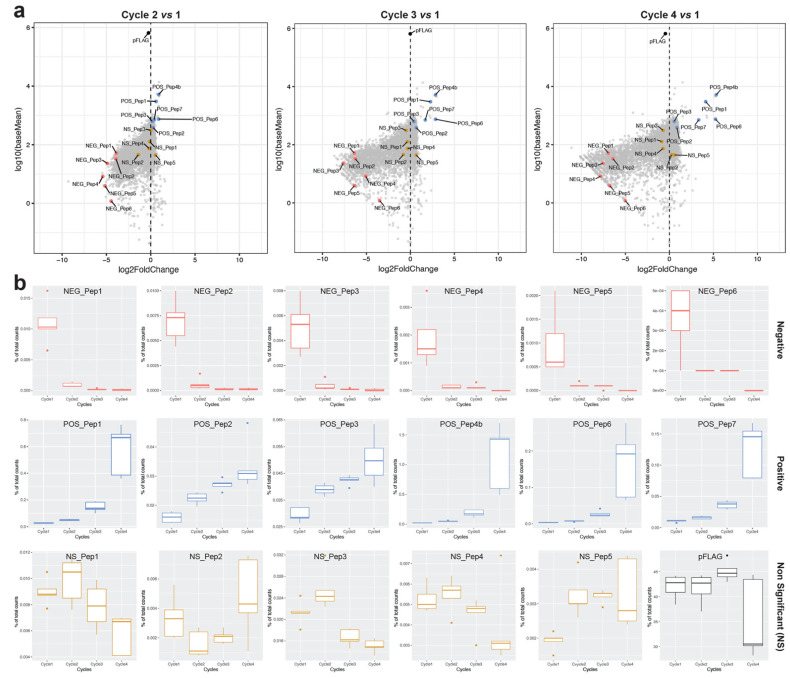
Performance characteristics of the candidate clones in the bulk experiment. Data are based on the deep sequencing experiment described in [14]. (**a**) Frequency changes of candidate clones during the growth cycles of the random sequence library in the background of all analyzed clones. Plots are based on the statistical analysis of clone frequency changes with DESeq2 [18], comparing mean counts of clones in the starting library (cycle 1) versus fold-changes at cycles 2, 3 and 4 (4 serial passages) for the deep sequencing experiment in [14] and its reanalysis in [18]. The *y*-axis shows the average read counts across the replicates (log10 (baseMean)); the *x*-axis is the relative change in clone frequency (log2FoldChange) at each cycle versus the counts at cycle 1. Candidates that are the focus of this study are highlighted with colored dots. We categorise the candidates into three classes—NEG_Pep = negative (decreased in frequency—padj < 0.05 and negative log2FoldChange value), NS_Pep = neutral (no significant change in frequency—padj > 0.05), POS_Pep = positive (increased frequency—padj < 0.05 and positive log2FoldChange values), and pFLAG = empty plasmid (black dot). Note that the overall effect of the empty plasmid in this experiment is within the range of the effects of the neutral peptides, but this can vary somewhat between experiments [18]. (**b**) Individual frequency trajectories of the candidate clones during the four cycles of growth. Boxplots show medians with lower and upper hinges corresponding to the 1st and 3rd quartiles.

**Figure 2 genes-13-00053-f002:**
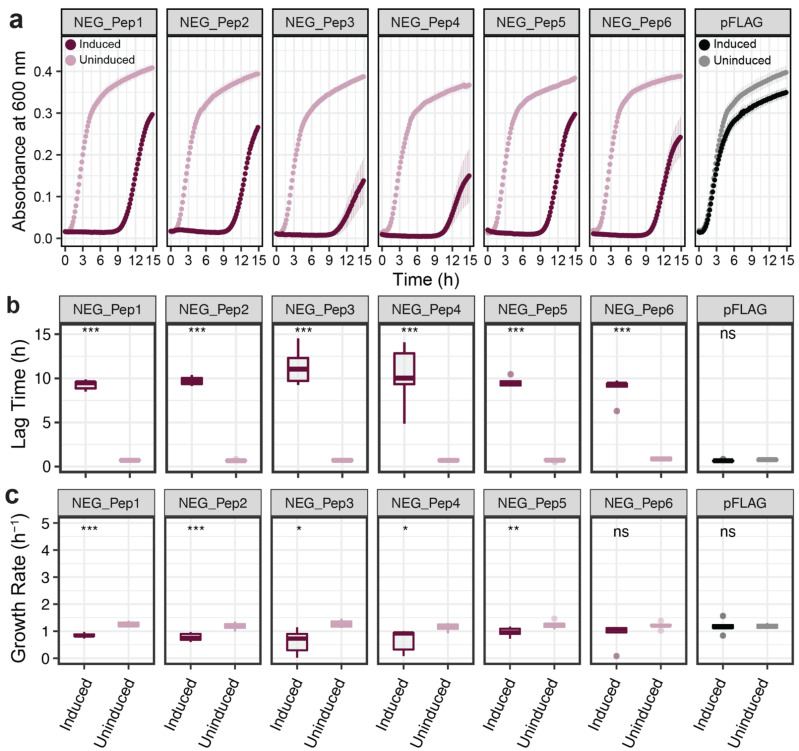
Growth rate comparisons for NEG_Pep clones compared to empty vector. (**a**) Growth curves without IPTG induction are depicted as light colors and with IPTG induction are in dark colors. OD_600_ was recorded every 10 min with at least five replicates for each clone and condition. Dots represent means and whiskers, showing the standard error of the mean (SEM) across replicates. (**b**) Lag time comparisons between the six clones, measured as the time from the start of the experiment until the start of the exponential growth. (**c**) Growth rate comparisons between the six clones after the end of the lag phase. Boxplots show medians with lower and upper hinges corresponding to the 1st and 3rd quartiles. Student’s *t*-test was performed with *p*-values as follows: *** = *p* < 0.001, ** = *p* < 0.01, * = *p* < 0.05, ns = *p* > 0.05.

**Figure 3 genes-13-00053-f003:**
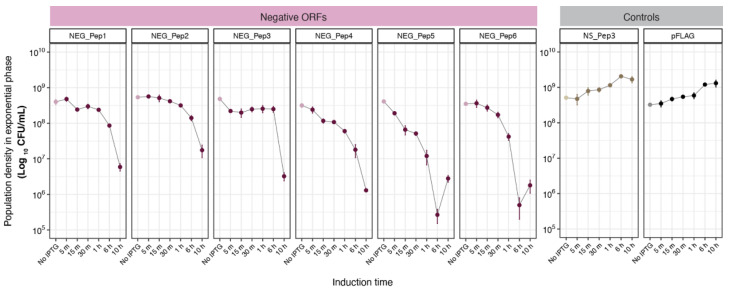
Test for the viability of cells after NEG_Pep induction. Colony counts (CFU/mL) of bacteria in the log phase, expressing either negative, non-significant or pFLAG peptides at different times after IPTG addition, show a decline in the NEG_Pep strains. Growth of the NS_Pep and pFLAG controls, on the other hand, is not affected by IPTG induction. Each panel’s light color dots represent growth under no induction, followed by IPTG addition (darker dots). Three replicates were done for each time point; whiskers in each dot represent SD.

**Figure 4 genes-13-00053-f004:**
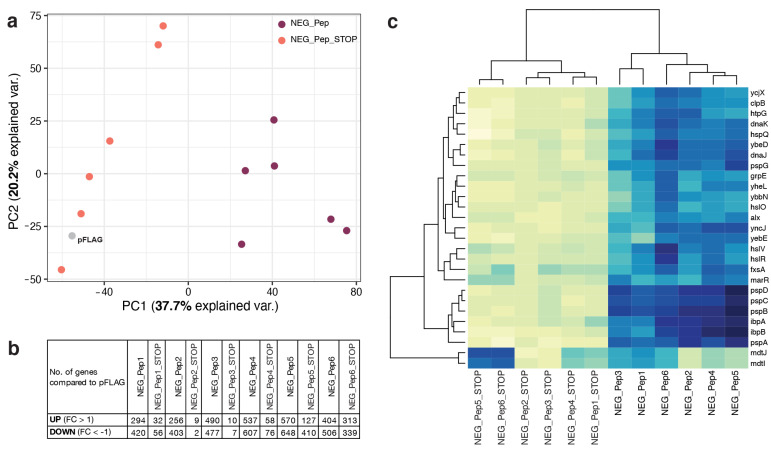
Transcriptomes of host cells expressing NEG_Pep clones show differential up-regulation of genes compared to their stop codon versions. (**a**) Principal component analysis (PCA); dots represent average expression from three replicates for each sample. (**b**) Numbers of genes above and below log2 fold change (FC) of 1 and −1 are shown here respectively. (**c**) Top highly expressed genes in all NEG_Peps are represented in the heatmap with dendrograms highlighting the differential expression in their corresponding stop codon versions. The *x*-axis shows the sample names and *y*-axis lists the top expressed genes (log2 FC > 4) from the expression analysis.

**Figure 5 genes-13-00053-f005:**
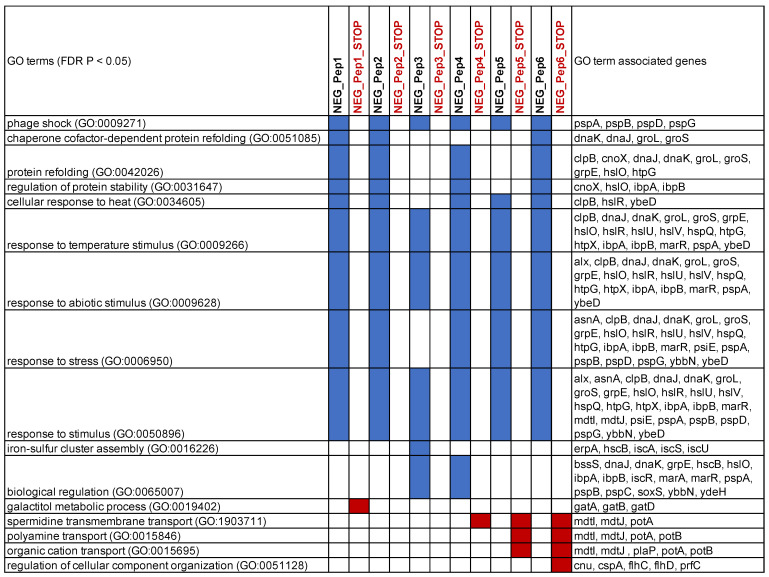
Major GO enrichment categories for the genes with the highest fold change in cells expressing NEG_Pep clones. Categories are listed with FDR *p* < 0.05; redundant categories were removed. GO terms were extracted using the GO enrichment analysis tool using Panther version 16.0. Blue and red marks indicate that the respective GO categories were found for the NEG_Pep listed in the respective column. The GO term-associated gene list is cumulative for all clones that show the respective GO term.

**Figure 6 genes-13-00053-f006:**
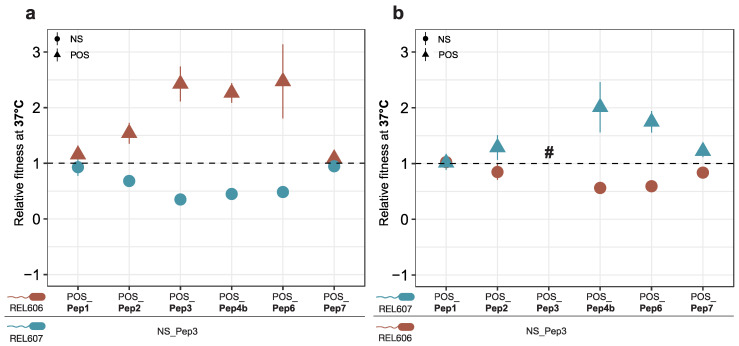
Competitive fitness analysis for six POS_Pep clones. The respective plasmids were cloned into the *E. coli* B REL606 and REL607 ancestor backgrounds that produce red and pink colonies, respectively, on TA indicator plates (see Methods). The competitor in each experiment was the NS_Pep3 clone. (**a**,**b**) represent the experiments for swapped backgrounds to ensure that there are no background-specific effects. The symbol # represents missing data. At least four replicates were used for each experiment. Whiskers represent the SEM.

**Table 1 genes-13-00053-t001:** Transcriptomic responses of cells expressing POS_Pep clones.

Clone	Down *	Up *	GO Enrichment Term	Enriched Genes
POS_Pep1	29	74	GO: 0015834 peptidoglycan-associated peptide transport	oppD, oppB, oppC, oppF
POS_Pep2	4	9	none	NA
POS_Pep3	4	20	GO: 0051454 intracellular pH elevation	gadA, gadB, gadC
POS_Pep4b	22	5	GO: 0046392 galactarate catabolic process	garK, garR, garL, garD
POS_Pep6	17	5	none	NA
POS_Pep7	8	5	GO: 1990451 cellular stress response to acidic pH	hdeA, hdeB

* Number of genes with log_2_ fold changes <−1 (down) and >1 (up).

## Data Availability

Scripts and data for the microarray analysis are provided at https://github.com/DevikaBhave/RandPep.git (last accessed on 21 December 2021).

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
