# Peer review of "Effects of the Expression of Random Sequence Clones on Growth and Transcriptome Regulation in Escherichia coli"

_genes, 2021, doi:10.3390/genes13010053_

Round 1
Reviewer 1 Report
This manuscript is nicely written and easy to understand. Below I have listed a few points that raised concern (listed as they occur in the manuscript) that I would like the authors to address.
Lines 72 onwards. Saying that ORFs that do not cause a fitness decrement could “easily become subject to positive selection” is a bit of a stretch. Firstly, that there are so few known cases where this occurred is a strong indication that it is not so easy. (1) How do the authors envision this happening? through point mutations? Not likely, since most mutation are deleterious. (2) How do the authors envision these random sequences getting into the genome, and why would it be maintained (since bacteria seem not to accumulate munch extra DNA. Since this process can be viewed as tenuous, it needs to be reinforced because it seems to be the rationale of the entire study.
Line 112. Perhaps the authors mean 500 µl of the pre-culture, not 500 ml. First paragraph of Results. Even one sentence explaining how the positive, negative and non-significant peptides were chosen would be useful (e.g., was it based on any particular sequence characteristics?) It is not immediately clear what is meant by the clones “represent[ing] a range of log2fold changes and representations in the library” (for example, in Figure 1(a), it seems like the positive clones all represent log baseMean values > 2). Such a clarification would be especially useful for the non-significant peptides, since one would expect them to constitute the largest category. Also, although clone sequences are given in the supplementary information, I think it would be useful to mention the length of POS_Pep4b, 6 and 7 in the text because of the stop codon
Line 294: “We determined viable cell numbers during the lag phase”. Wasn;t the experiment in Figure 3 performed in log phase? Also, it would be useful to understand how the results of the experiment in Figure 3 explain that of Figure 2. In Figure 2, cells are starting to recover after 9 or 10 hours; but in Figure 3 growth at this point is at the lowest. Is this due to the rate of IPTG induction at different growth stages? Whatever the reason, some explanation to this effect would be useful to connect the two sets of experiments
Lines 315-316: The comparison between the peptide-driven delay and RNA-driven delay is mentioned in the discussion, but I think it should be mentioned here as well, since the immediate next section assays the comparative effects of peptide vs RNA expression on the transcriptome. Transcriptome analysis: The overall takeaway from the transcriptome and gene enrichment analysis seems to be that negative clones cause a generic stress response, whereas positive ones do not. Please explain how and whether the generic stress response is triggered by effects specific to the negative clones, or if expression of neutral/non-significant peptides would have a similar effect as well. (it could be that “positive” clones that anomalous ones since they do not cause a stress response upon expression.
Lines 310 onwards. The analysis of the clones causing deleterious effects would be strengthened if the authors, in addition to reporting the GO categories of the potential protein, reported the extent of sequence similarity of each RNA to that present in the genomes. Note that the effects need not rely on overall sequence similarity since even a short stretch can have disruptive effects.
Lines 464-465: Perhaps I missed something, but I do not view this as being substantiated by the work presented in this paper
Line 484-485: Similarly, is the 10% figure substantiated in this paper? Also, Figure 6 shows that bulk experiment advantages might not always be translated to fitness advantages in head-to-head competition experiments, so could it not be the case that some of these positive growth effects can be attributable to vector effects? Also, f the argument is that somehow new peptide insertions occurred in the LTEE, they would have fixed quicker than point mutations, then that requires some data to be substantiated as well, since there are differences between growth experiments over relatively few generations vs. the long-term evolution experiment.
Reviewer 2 Report
# General remarks
The paper is well written and pleasant to read.
The data availability statement is incomplete and does not mention the code
that generated the various analyses. It is of crucial importance that all code is
deposited on a platform such github or gitlab in order to ensure reproducibility.
The ERC, that funded these research embraces FAIR data principles.
Moreover, there are two instances of _data not shown_ in the paper.
These data should appear as supplementary data or be deposited on a platform
such as Zenodo.
# 1 Introduction
The introduction is very concise and well written and allows the reader to
get a good general idea of the topic at hand without too much details.
# 2 Materials and Methods:
- l.153: The term `NEG_peps` should be defined before use
(it is defined later at l.269 in the legend of Figure 1).
# 3 Results:
## 3.1.1
- l.252:The `log2fold changes` seems to be in terms of frequency at some point
compared to the initial point as explained in the legend of Figure 1 it
should be clearly stated.
- About Figure 1, I would like to know if the authors have any idea why the
distribution of `% of total counts` display such differences between the
cycles (for example `POS_Pep6 -- Cycle 4` vs `Cycle 3`)?
- l.282: I assume the authors are reffering to Figure 2C and not Figure 2B
which shows `Growth Rate` and not `Lag Time`.
## 3.1.2
- Figure 5: I think the authors could make the figure smaller and more imediate
for the reader by pruning the GO terms presented.
`response to stimulus (GO:0050896)` and `biological regulation (GO:0065007)`
for example are not very informative
## 3.1.4
- In the Table 1, the authors present an analysis that mirrors the one done in
3.1.2 but is presented in a very succint table instead of a heatmap.
The comparison between transcriptomic response between NEG and POS
require in my opinion the presented figure to have the same kind of layout.
## 4
- l.441 to l.443 I'm not sure to understand the sentence (also instead of although ?),
could the author explain its meaning in the response and/or rephrase it in the paper ?
-
